# Tumor-Associated Macrophage Targeting of Nanomedicines in Cancer Therapy

**DOI:** 10.3390/pharmaceutics16010061

**Published:** 2023-12-29

**Authors:** Xuejia Kang, Yongzhuo Huang, Huiyuan Wang, Sanika Jadhav, Zongliang Yue, Amit K. Tiwari, R. Jayachandra Babu

**Affiliations:** 1Department of Drug Discovery and Development, Harrison College of Pharmacy, Auburn University, Auburn, AL 36849, USA; xzk0004@auburn.edu; 2Materials Research and Education Center, Materials Engineering, Department of Mechanical Engineering, Auburn University, Auburn, AL 36849, USA; 3Zhongshan Institute for Drug Discovery, Shanghai Institute of Materia Medica, Chinese Academy of Sciences, Guangzhou 528400, China; yzhuang@simm.ac.cn; 4Shanghai Institute of Materia Medica, Chinese Academy of Sciences, Shanghai 201203, China; wanghuiyuan@simm.ac.cn; 5Department of Pharmaceutical Sciences and Experimental Therapeutics, College of Pharmacy, University of Iowa, Iowa City, IA 52242, USA; sanika-jadhav@uiowa.edu; 6Department of Health Outcome and Research Policy, Harrison School of Pharmacy, Auburn University, Auburn, AL 36849, USA; zzy0065@auburn.edu; 7Department of Pharmaceutical Sciences, College of Pharmacy, University of Arkansas of Medical Sciences, Little Rock, AR 72205, USA; atiwari@uams.edu

**Keywords:** tumor-associated macrophage, suppressive immune environment, tumor proliferation and metastasis, nanomedicine, targeted delivery systems

## Abstract

The tumor microenvironment (TME) is pivotal in tumor growth and metastasis, aligning with the “Seed and Soil” theory. Within the TME, tumor-associated macrophages (TAMs) play a central role, profoundly influencing tumor progression. Strategies targeting TAMs have surfaced as potential therapeutic avenues, encompassing interventions to block TAM recruitment, eliminate TAMs, reprogram M2 TAMs, or bolster their phagocytic capabilities via specific pathways. Nanomaterials including inorganic materials, organic materials for small molecules and large molecules stand at the forefront, presenting significant opportunities for precise targeting and modulation of TAMs to enhance therapeutic efficacy in cancer treatment. This review provides an overview of the progress in designing nanoparticles for interacting with and influencing the TAMs as a significant strategy in cancer therapy. This comprehensive review presents the role of TAMs in the TME and various targeting strategies as a promising frontier in the ever-evolving field of cancer therapy. The current trends and challenges associated with TAM-based therapy in cancer are presented.

## 1. Introduction

The “Seed and Soil” theory of cancer proposes that successful cancer growth and metastasis depend not only on the intrinsic characteristics of cancer cells (the “seed”) but also on the specific tumor microenvironment (TME) (the “soil”) [1]. The TME is a complex system consisting of immune cells, stromal cells, and heterogeneous cancer cells [2]. Among the cellular components, macrophages play a crucial role and can account for 30–50% of the tumor mass [3]. Macrophages (Mø) are recruited to the TME chemo-attractants like chemokine (C-C motif ligand) 2 and 5 (CCL2, CCL5) [4]. The presence of anti-inflammatory cytokines such as IL-4, IL-10, colony-stimulating factor (CSF), and TGF-β promotes the differentiation of recruited Mø into TAMs, which, in turn, enhance tumor cell growth [5,6]. The macrophages in the TME are termed tumor-associated macrophages (TAM), which include M1 and M2 phenotypes [7,8]. M1 macrophages have anti-tumor properties, while M2 macrophages are pro-tumoral in nature [7,8].

High levels of infiltration of M2 TAMs within tumors have been linked to unfavorable survival outcomes [9]. M2 TAMs provide trophic support to cancer cells by supplying growth factors and nutrients, contributing to cancer cell survival and proliferation. TAMs also facilitate tumor invasion and metastasis through the degradation of the extracellular matrix via the secretion of enzymes like MMPs [10,11]. Molecules including cytokines such as interleukin-10 (IL-10) and transforming growth factor-beta (TGF-β) inhibit the activity of cytotoxic immune cells and promote regulatory immune cell populations [12,13]. Additionally, TAMs secrete factors that promote angiogenesis, such as vascular endothelial growth factor (VEGF) and matrix metalloproteinases (MMPs), facilitating the formation of new blood vessels to nourish the growing tumor [14,15]. As a feedback, tumor cells release factors like IL-12, IL-4, hypoxia-inducible factor (HIF)-1α, and HIF-2α to sustain the M2 TAM phenotype [16,17]. Thus, the targeting of TAMs (improvement of cancer “soil”) is beneficial in the cancer treatment.

The strategy for targeting TAMs involves blocking chemokines and growth factors that attract TAMs to the tumor site [18], elimination of M2 TAMs, or impairing TAM function with therapeutics [19], Additionally, TAMs are often polarized into two main subtypes: M1 (anti-tumoral) and M2 (pro-tumoral). Reprogramming M2 TAMs into anti-tumor M1-TAMs is a promising strategy [7,8]. Recently, boosting TAMs’ phagocytic capacity with immunotherapeutic strategies has raised attention [20]. The CD47-SIRPα pathway on tumor cells prevents phagocytosis by binding to macrophage SIRPα, serving as a “do not eat me” signal. Inhibiting this pathway with CD47 antibodies and engineered SIRPα proteins improve macrophage phagocytosis, promoting CD8 T cell activity and suppressing Treg cells [21,22]. The CD-Siglec axis involves CD24-Siglec interactions, inhibiting innate immune reactions [23,24]. Blockade of Siglec-15 with the monoclonal antibody NC318 enhances CD8+ T and NK cell infiltration [25]. Targeting MHC class I/LILRB1 interaction and LILRB2 facilitates macrophage activation. PD-L1/L2 inhibition, along with CD47 and PD-L1 antibodies [26], synergistically enhances anti-tumor immunity, offering potential breakthroughs in cancer therapy [25].

After selecting the appropriate therapeutics, the design of appropriate drug delivery systems to overcome barriers are important. Given the enhanced permeability and retention EPR effects, nanoparticles are emerging as powerful tools for targeting and regulating TAMs [27]. Nanoparticles can be tailored to have TAM-favored size (10~500 nm) and shapes that mimic pathogens, making Nps more attractive to macrophages’ uptake [27]. To further effectively target TAMs, active targeting strategies involving modifying nanomedicines with specific ligands that are unique to M2 TAM macrophages have surfaced as a revenue [28,29]. This review provides readers with comprehensive knowledge of Nps-based TAM strategies in cancer therapy by concentrating on strategies involving nanoparticles or nanomedicines targeting TAMs.

While significant progress has been made in the field of immunotherapy, the effectiveness of these treatments in combating solid tumors has fallen short of expectations in clinical applications [30]. The complexity and heterogeneity of TAM populations, particularly in humans, pose challenges for developing targeted therapies, demanding robust biomarkers for accurate patient stratification. Efficient delivery and specific targeting of TAM therapies face hurdles due to the varied distribution of TAMs within the tumor microenvironment. Modulating TAM activity can have systemic effects, emphasizing the need to understand potential side effects and toxicities. Addressing these challenges requires integrated knowledge in TAM biology, biomarker identification, and delivery systems. This review was compiled from the published articles through November 2023 from various databases such as PubMed, Google Scholar, and Web of Science. We searched the keywords including “Tumor-associated macrophage, Suppressive immune environment, Tumor proliferation and metastasis, Nanomedicine, Targeted delivery systems” and extracted the key points and presented this review.

## 2. The Role of TAMs in Tumor Immune Microenvironment (TIME)

TAMs are crucial components of the tumor immune microenvironment (TIME). Two classic macrophages are the M1 and M2 phenotypes; among them, the M1 macrophage represents the beneficial anti-tumor macrophage, while the M2 macrophage represents the tumor-promoting macrophage. In the tumor region, the recruited monocyte is easily differentiated into M2-type TAMs due to the abundance of Th2 cytokines (IL-4, IL-10, IL-13), limiting antigen presentation and promoting immunosuppression [31]. This hinders the immune system’s ability to recognize tumor cells. The interaction of TAMs with other cells in the tumor microenvironment facilitates the immunosuppressive microenvironment [32].

On one side, M2 TAM favors immune suppressive cells. Cancer-associated fibroblasts (CAFs) are located near TAMs, jointly forming the physical barrier around tumor cells that hinders immune cell and therapeutic agent access [33]. Additionally, TAMs and CAFs mutually stimulate each other’s functions, promoting immunosuppression and fibrosis [34,35]. TAMs also interact with T regulatory cells (Tregs). TGF-β released by TAMs induces Treg differentiation and function [36]. Conversely, Tregs suppress M1 TAMs and enhance M2 TAMs, impacting CD8+ T cell function [37,38,39]. Thus, M2 TAMs and Treg are generally regarded as immune suppressive cells in TME [37,38].

On the other hand, TAMs impair the function of immune active cells. TAMs inhibit natural killer (NK) cell activation by releasing HLA molecules and TGF-β [39,40]. Additionally, TAMs impair dendritic cell (DC) function via IL-6, nitric-oxide synthase 2 (NOS2), cyclooxygenase-2 (COX2) [41,42], VEGF [43,44,45], prostaglandin E2 (PGE2) [46,47], and TGF-β [48,49]. M2 TAMs impair cytotoxic T cell (CTL) function through the production of arginase 1 (Arg-1) and IL-10, affecting polyamines and proline metabolism [50,51]. TAMs also inhibit IL-12 production by DCs, limiting the anti-tumor response of CD8+ T cells [52,53]. TAMs express indoleamine-2,3-deoxygenase (IDO), which produces kynurenine and impairs the function of T cells as well as NK cells; the IDO also supports the function of T regulatory (Treg) cells [41,42,43,54]. These interactions reflect the critical role of TAMs in shaping the immune response in the tumor microenvironment (Figure 1).

## 3. TAMs and Metastasis

### 3.1. TAMs, Angiogenesis, and Lymph-Angiogenesis

Angiogenesis is a crucial process in tumor progression, involving basement membrane degradation, activation and proliferation of endothelial cells (ECs), and the formation of capillary tubes and tube-like structures [44,45]. Hypoxia serves as a key driving factor for angiogenesis [46], and TAMs can sense tumor hypoxia and produce various angiogenic factors, including hypoxia-inducible factors (HIFs) [46], TGF-α, TGF-β, [47], vascular endothelial growth factor (VEGF) [48,49], and matrix metalloproteinases [55,56]. Knockout of the HIF-1α gene in TAMs results in a reprogramming of M2 TAMs to M1 TAMs and attenuates their pro-angiogenic abilities [57]. Additionally, TAMs promote tumor cell aerobic glycolysis, leading to increased lactate levels that induce angiogenesis [58]. Angiopoietin-2 (Ang-2) recruits TIE2 receptor-expressing monocytes (TEM), facilitating angiogenesis [59,60,61].

Lymph-angiogenesis is critical for the formation of tumor lymphatic vessels [62]. TAMs produce pro-lymphangiogenic factors such as lymphatic vessel endothelial hyaluronan receptor 1 (LYVE-1) and a glycoprotein YKL-40 to support tumor lymph-angiogenesis [63]. LYVE-1, a homolog of CD44 and a marker specific to lymphatic vessels [64], has been identified in tissue-resident macrophages. LYVE-1 positive M2 TAM is crucial for the development of dense vascular networks and supporting angiogenesis [65]. Additionally, YKL-40, a chitinase-like protein, could activate FAK-MAPK signaling and increase the expression of VEGFR1 and VEGFR2 [66]. Moreover, TAMs indirectly regulate lymph-angiogenesis through the production of enzymes that remodel the matrix and activate growth factors [67].

### 3.2. Metastatic Colonization and Survival

TAMs play a critical role in cancer cell migration during metastasis through autocrine loops [68,69,70]. TAMs also contribute to the formation of pre-metastatic supportive niches (PMNs) by responding to tumor-released CCL2, recruiting macrophages to the metastatic site [71]. Within the metastatic niche, these macrophages are referred to as metastasis-associated macrophages (MAMs) [72,73]. MAMs further produce cathepsin S and VEGF-A, which increase vessel permeability and promote cancer cell extravasation [74]. Moreover, macrophage-generated granulin activates resident stellate cells, leading to their transformation into myofibroblasts that produce fibronectin, feriostin, and collagen, supporting cancer cell metastasis [75]. Additionally, TAMs promote the intravasation of tumor cells into the host vasculature through the secretion of EGF and CSF-1 [76].

Macrophages play a crucial role in promoting metastatic tumor cell survival [77]. They produce integrin α4, which interacts with VCAM1 on tumor cells, leading to enhanced survival via PI3K/Akt signaling [78]. Additionally, macrophages bind to fibrin complexes on tumor cell-associated platelets, facilitating their survival during the initial stage of metastatic colonization [79]. Furthermore, TAMs promote tumor cell invasion by mediating the degradation of the extracellular matrix through the involvement of cathepsins and matrix metalloproteinases (MMPs) such as MMP7, MMP2, and MMP9 [80]. Macrophages are also implicated in driving epithelial-mesenchymal transition (EMT) through factors like transforming growth factor-β (TGF-β) and IL-8 [81] (Figure 2).

## 4. TAM and Therapeutic Resistance

TAMs play pivotal roles in therapeutic resistance in various cancers, making them important targets to improve treatment efficacy [82]. Chemotherapeutic regents achieved success for decades; however, TAM impairs the efficacy of chemotherapy via factors including CSF-1 [83], CCL2/CCR2+ axis [84], and macrophage inhibitory factor (MIF) [85]. In addition, the mechanism of TAM is involved in the drug resistance including the activation of anti-apoptotic signals and the promotion of tumor-favorable Th17 response after the treatments with chemo-agents [86,87]. In addition, the TAMs also play a pivotal role in connecting epithelial-mesenchymal transition (EMT) with therapeutic resistance [88]. The interplay between cytokines produced by TAMs and cancer cells fosters EMT and the acquisition of stem cell-like properties [88]. Cancer cells undergoing EMT exhibit characteristics such as enhanced drug expulsion and resistance to apoptosis, rendering them highly resilient to chemotherapy drugs [89].

In molecular target therapy, TAMs have been implicated in resistance to EGFR-TKIs in lung cancer [90]. Combination therapy targeting both EGFR and TAMs holds promise [91]. TAMs can also contribute to resistance to radiotherapy [92,93]. Strategies like CSF-1R inhibition and macrophage depletion have been explored to enhance radiotherapy efficacy [94,95]. In antibody-based therapy, TAMs can compromise the effects of drugs like trastuzumab and immune checkpoint inhibitors [96,97]. Depleting TAMs may improve the response to such therapies. TAMs also play a role in resistance to anti-angiogenic therapy by infiltrating and producing tumor-promoting factors [98]. It was found that the selective elimination of M2 TAMs has shown potential in enhancing the efficacy of anti-angiogenic therapies [99] (Figure 2).

## 5. Therapeutic Strategies

As previously mentioned, the presence of TAMs in TME is correlated with cancer progression and treatments. Therefore, the therapeutic strategies of targeting macrophages are significant (Figure 3).

### 5.1. Typical Conventional Therapeutics

Because of the abundant literature summarizing the traditional therapeutics, we will only briefly cover the conventional strategies. The conventional therapeutics focus on TAM based on the below three aspects: (a) Inhibition of TAM recruitment and differentiation using inhibitors for pathways including CSF-1/CSF-1R [100,101], CCL2/CCR2, and CCL5/CCR5 pathways [102,103]. (b) Depletion of TAM and impairment of TAM function with various agents, including trabectedin and amphotericin B [104,105] and bisphosphonates [106,107,108]. Additionally, the mannosylated engineered trichosanthin-legumain (MTL) vaccine selectively depletes M2 TAMs, which in turn activates CTLs, benefiting the treatment of breast cancer [105]. (c) Reprogramming M2 TAMs using TLR agonists like resiquimod [109] and CpG ODNs [110] to trigger M2-M1 TAM repolarization. Inhibitors of STAT3 and STAT6, such as FLLL32 and corosolic acid, have also been found effective in reprogramming TAM [111,112,113]. Additionally, PI3K inhibitors (e.g., IPI-549) [114] and HDAC inhibitors (e.g., TMP195) [115,116] offer potential strategies for TAM remodeling (Figure 3). For more compounds associated with the above therapeutic targets, refer to Table 1.

The therapeutic significance and approaches to targeting TAMs have been discussed in the above sections. Based on the increased number of publications in recent years, targeting TAMs is gaining high significance and appears to be a more practical approach in cancer therapy. Strategies focused on TAMs have made successful progress. However, there are obstacles. First, cessation of M2 TAM therapeutics may cause the rebound of M2 TAM. Second, the off-target side effects because of mixed sub-type in TIME and interaction with healthy macrophages result in failed effects. Researchers are gradually paying attention to the biological functions in terms of macrophage recovery. Thus, in the following part, the efforts in investigating the TAM phagocytosis function are summarized.

### 5.2. Recovery of Phagocytosis of TAM

In the tumor’s initial stage, the macrophage has the phagocytic ability towards cancer cells. However, cancer cells program the beneficial macrophage to a pro-tumor M2 macrophage that highly expresses the immune markers. Thus, the use of an immune checkpoint blockade is a promising approach in the recovery phagocytosis ability of macrophages and facilitates the T cell function in the body, in which the modulation of certain signaling pathways is crucial (Figure 3).

**CD47-SIRPα pathway:** CD47 is an inhibitory ligand on tumor cells that binds to the signal-regulatory protein alpha (SIRPα) on macrophages, acting as a “do not eat me” signal to prevent phagocytosis [148]. Blocking the CD47-SIRPα signaling pathway using CD47 antibodies (such as Hu5F9-G4 and CC-90002) [21,22], engineered SIRPα proteins (e.g., ALX148) [21], or SIRPα-Fc fusion proteins (e.g., TTI-621) [149] enhances macrophage phagocytosis. The increased phagocytosis promotes CD8 T cell activity and suppresses Treg cells [31,150,151]. Additionally, the blockage of interaction between tumor cells and TAM might not be enough to fully convert the “soil” environment. Combination therapies that target CD47 signaling, along with the blockade of TAM recruitment using CSF-1R inhibitors, have demonstrated improved anticancer efficacy [152,153,154].

**CD-Siglec axis:** Small-cell lung carcinoma cluster 4 antigen (CD24) expressed on tumor cells interact with sialic acid-binding immunoglobulin-like lectin 10 (Siglec-10) expressed on TAMs [23,24]. Through the Src homology region, two domain-containing phosphatases (SHP-1 and/or SHP-2) mediated inhibitory signals were generated, suppressing innate immune cells’ reaction against tumor cells [155,156]. In addition to siglec-10, further studies indicated that siglec-15 and siglec-8 are also involved in the anti-phagocytosis process [157,158,159]. Similarly, the blockade of Siglec-15 enhanced the infiltration of CD8+ T and NK cells into tumors, resulting in reduced tumor burden and prolonged survival in mice [157,158,159]. The monoclonal antibody NC318, which targets Siglec-15, is currently undergoing clinical trials. This antibody offers hope to individuals who do not respond to PD-1/PD-L1 therapy [25]. The development of this antibody represents a potential breakthrough in the treatment of non-responders, bringing new possibilities for improving outcomes in cancer immunotherapy [160,161]. Antibody-toxin conjugate can also be used to selectively deplete Siglec-8 positive immune-suppressive immune cells [162].

**Major histocompatibility (MHC) class I–LILRB1/LILRB2 signaling axis:** The expression of major histocompatibility complex class I (MHC I) on cancer cells hinders phagocytosis by interacting with the leukocyte immunoglobulin-like receptor (LILRB1) on macrophages [163]. The beta-2 macroglobulin subunit of MHC I (B2M) is a newly founded anti-phagocytic surface protein (“do not eat me” signal) [163]. B2M suppresses the phagocytic function of macrophages by interacting with its inhibitory receptor LILRB1 on TAMs, leading to compromised anti-tumor immunity [163]. Targeting the MHC class I/LILRB1 interaction to enhance TAM phagocytosis is a potential therapeutic approach to cancer treatment [163]. LILRB2, another member of the LILRB family, has been found to be expressed in various cell types, including monocytes and macrophages [164,165]. Therapeutic antibodies targeting LILRB2, which also interacts with HLA class I similar to LILRB1, have been shown to facilitate macrophage maturation and promote their pro-inflammatory activation [166,167]. Of note, it remains uncertain whether the promotion of phagocytosis activities by antagonizing LILRB2 occurs directly or indirectly through macrophage phenotypic changes [164,168]. While both LILRB1 and LILRB2 bind to HLA-I/MHC-I, it is yet to be determined if the interaction of LILRB2 acts as a phagocytosis checkpoint [169].

**PD-L1/L2 and PD-1:** The PD-1/PD-L1 interaction primarily occurs in the context of the adaptive immune response, involving T cells and other immune cells. Additionally, the expression of PD-L2 on M2 TAM affects the anti-tumor effects of Th2 T cells [170]. The inhibition of PDL2/PD1 with nivolumab benefits the recovery of Th2 T cell function [171]. Additionally, the use of antibodies against CD47 and PD-L1 was shown to exert synergistic anti-tumor immunity effects [172]. A ROS-responsive albumin-based system has been developed to sequentially release antibodies against CD47 and PD1, promoting M1 TAM differentiation and leading to an enhanced anti-tumor response [173].

### 5.3. Modality for Metabolism in TAM

Modulating the metabolism of TAMs is important because of the direct association with TAM polarization [174,175]. TAMs show elevated expression of glutamic pyruvic transaminase and glutamine synthetase involved in glutamine metabolism [176]. M2 TAM exhibits increased glutamine catabolism [176]. The enhanced utilization of glutamine by TAMs fuels the production of the chemokine C-C motif chemokine ligand-22 (CCL22) and contributes to the M2 TAM phenotype via the glycosylation of C-type lectin receptors like CD206 and CD301 [177]. Disulfiram modulates the metabolism in the glioma TME via glucose-glycolysis/folate-NADH-ATP metabolism axis; thus, the anti-tumor M1 TAMs become predominant [178]. Additionally, inhibiting oxidative metabolism causes the M2-To-M1 phenotype [179]. Shikonin, a natural product, acts as a regulator of TAM via the regulation of colorectal cancer [180]. The intricate interplay between tumor metabolism, angiogenesis, and immunity constitutes a complex network [181]. Cancer cells undergo a metabolic reprogramming of glycolysis, which can be mediated by mTOR. Thus, mTOR inhibitor can be a therapeutic for cancer. A combinational therapy of rapamycin and regorafenib (inhibitor for angiogenesis) regulates the metabolic activity in the TME. This TME reprogramming resulted in reduced proliferation of cancer cells and a decrease in the production of lactic acid that possesses immunosuppressive properties and can also stimulate angiogenesis within the tumor microenvironment [182] (Figure 4).

## 6. Nanotechnology-Based TAM Imaging and Therapeutic Delivery System

Nanoparticles enhance the drug delivery in tumor sites via the Enhanced Permeability and Retention (EPR) effect, which is known to be a well-known strategy. The abnormal vasculature in tumors is characterized by increased permeability and compromised lymphatic drainage, which allows nano-size drug carriers to selectively accumulate in the tumor tissues. Therefore, nano carriers reduce off-target effects on healthy tissues. In this review, we explore the use of nanoparticles for targeting TAMs in the TME.

### 6.1. Nanoparticle for Imaging TAM

The development of imaging agents for the specific visualization of TAMs in vivo involves various innovative approaches. Molecular imaging techniques like magnetic resonance imaging (MRI) and Positron Emission Tomography (PET) offer non-invasive means to observe and monitor the behavior of TAMs within tumors. This is achieved by tracking and quantifying the uptake of MRI-visible nanoparticles or PET-visible radiotracers that are specific to TAMs.

#### 6.1.1. MRI-Visible Nanoparticles

Iron-based nanoprobes or gadolinium-based nanoparticles can be used as MRI contrast agents [183]. A notable example is the development of PEG-b-AGE polymer-coated iron oxide nanoparticles engineered to target the mannose receptor found on M2-like macrophages. This nano platform exhibits robust imaging capabilities for M2-like macrophages within the tumor microenvironment [183]. Additionally, to monitor the imaging of glioblastoma (GBM) TAMs, Runze Yang and colleagues recently showed that the use of iron oxide nanoparticles (USPIOs) on their own in MRI effectively tracked the monocyte-macrophage system [184].

Sulfated dextran-coated iron oxide nanoparticles, referred to as SDIO, have demonstrated the ability to target the macrophage scavenger receptor A (SR-A) and exhibit good retention within macrophages [185].

Another approach involves superparamagnetic iron oxide (SPIO) nanoparticles functionalized with an M2 macrophage-targeting peptide, empowering them with M2 macrophage-targeting properties and MRI capabilities. In vitro and in vivo experiments have validated their efficiency as both an imaging agent and an M2 macrophage-targeting tool [186]. In a study by Zhu et al. [187], they designed nanoprobes that utilized Erbium (Er)-based near-infrared IIb (NIR-IIb) fluorescence specifically targeting M2-type TAMs in orthotopic glioblastoma. Weissleder et al. synthesized a library of nanoparticles and found that the dextran-coated iron oxide nanoparticles (CLIO680 and AMTA680) exhibit a high affinity for TAMs [188].

#### 6.1.2. PET-Visible Radiotracers

PET is a functional imaging technique that measures metabolic activity within the body, in which the polarization, reduction, and recruitment (inhibition) of TAMs can be assessed [189]. Blykers et al. (2015) have developed 18F-labeled single-domain antibody fragments (sdAbs) derived from camelids with the specific aim of targeting the mannose receptor in M2-type TAMs [190]. In addition, a 68Ga-labeled single-domain antibody fragment (sdAb) targeting the MMR (Mannose Receptor) has been developed to evaluate the presence of pro-tumor TAMs [191]. The preclinical testing of [68Ga]Ga-NOTA-anti-MMR-sdAb demonstrated significant and specific uptake of this tracer in MMR-expressing TAMs and organs without any observed toxic effects. As a result of these promising findings, [68Ga]Ga-NOTA-anti-MMR-sdAb is now poised for advancement into a phase I clinical trial [191]. Additionally, serum albumin modified with mannose molecules and radionuclide-labeled nanobodies can be used as PET nanoprobes [192,193].

In 2021, two radiolabeled arginase inhibitors, namely 18F-FMARS and 18F-FBMARS, have been created using derivatives of α-substituted-2-amino-6-boronohexanoic acid to image TAMs in a prostate cancer xenograft model [194].

Specially engineered ultrasmall copper nanoparticles (Cu@CuOx) that target CCR2 have been designed as nanovesicles, by which accurate detection in pancreatic ductal adenocarcinoma (PDAC) mouse models was achieved [195]. The 64Cu-labeled nanovehicle nanoparticles demonstrate minimal in vivo toxicity [195]. However, PET tracers that target receptors found on TAMs were carried out in preclinical animal models and with only a limited number of tracers tested in human patients [196].

### 6.2. Therapeutic Nanoparticle for TAM

#### 6.2.1. Inorganic Nanoparticles

Gold materials (nanoparticles or nanocage) have been implicated in anti-tumor immunotherapy, and the immunomodulatory effects of AuNC are associated with either the reprogramming of TAM [197] or the depletion of M2 TAM [198]. The behind mechanism is associated with the heat and the heat-induced TME improvement [199]. In a study, Zhang et al. devised a strategy for in situ vaccination, employing gold nanocages (AuNC) that harness photothermal effects, along with an adjuvant and a PD-L1 inhibitor [197,200]. In addition, AuNCs resulted in autophagy intervention and thus contributed to TAM inhibition [201]. Furthermore, immune responses can be triggered by the release of tumor antigens and cell debris resulting from localized destruction of cancer cells through photothermal effects [202,203]. To facilitate drug delivery, the hydrophobic domains of albumin were found to interact with the drug paclitaxel (PTX), creating an albumin corona that acted as a drug carrier for gold nanorods (AuNRs). The co-loading significantly modulated the TME, effectively inhibiting the polarization of M2 TAMs [198]. Graphdiyne oxide nanosheets polarize M2-TAM macrophages for melanoma immunotherapy [204]. The intraperitoneal graphdiyne oxide injection reduces tumor growth and activates cytotoxic T cells [204] (Figure 5).

In addition, colorful and inorganic material can trigger photodynamic therapy [205,206], which involves the activation of a photosensitizer (PS) using light at a specific wavelength (λ) in the presence of molecular oxygen, resulting in the generation of singlet oxygen. Photodynamic therapy is able to modulate TME and affect TAM [206].

#### 6.2.2. Nanotechnology-Based-Large Molecules Therapeutics

Recently, large molecules, including protein, mRNA, siRNA, and miRNA therapeutics, have been used to reprogram TAMs [207]. Trichosanthin (TCS) has drawn much attention due to its newly founded immunomodulatory effects. A modified version of the drug trichosanthin called recombinant cell-penetrating trichosanthin (rTCS-LMWP) has been developed to repolarize TAM, remodel TME, and increase cytotoxic T cells, and suppress regulatory T cells [208] (Figure 6A). By delivering mRNAs encoding M1 macrophage, it is possible to enhance anti-tumor immunity [209]. In a study, researchers developed a targeted nanocarrier capable of delivering in vitro-transcribed mRNA encoding M1-polarizing transcription factors to TAMs. M2 TAM is typically predominant in TME. However, the nanocarrier specifically reprograms TAMs to adopt an M1 phenotype, unleashing anti-tumor immunity. The involved mRNAs are interferon regulatory factor 5 and kinase IKKβ (Figure 6B) [210]. Glioblastoma (GBM) has only a 6.8% five-year survival rate, and microglia and macrophages infiltrating the TME adopt a tumor-promoting phenotype (M2), which hinders the anti-tumor immune response in GBM. The researchers developed a virus-mimicking membrane-coated nanogel called Vir-Gel, which contained therapeutic miRNA to address this issue. This nanogel achieved M2-to-M1 alternation [211]. Additionally, in vivo experiments indicated that Vir-Gel effectively prolonged the circulation time of the therapeutic miRNA, allowing it to actively target tumors and exhibit excellent tumor inhibition efficacy (Figure 6C) [211]. In addition, targeted lipid-coated calcium phosphonate nanoparticles have also been designed for the delivery of microRNA (miRNA); this system responded to the acidic pH in TME [212].

In addition, mannose moiety facilitates enhanced internalization of TAMs towards miRNA. Consequently, the delivery system successfully downregulated pro-tumor factors, including IL-10, MMP9, and VEGF, but upregulated anti-tumor factors like IL-12 in TAMs [212]. Moreover, small interfering RNA (siRNA) has been utilized as a tool to reprogram M2 TAMs by specifically silencing target genes [213,214]. A lipid nanoparticle (LNP) composed primarily of the CL4H6 lipid was developed to deliver siRNA to TAMs. These optimized siRNA-loaded CL4H6-LNPs demonstrated efficient uptake by TAMs in the human tumor xenograft mice model, effectively silencing target genes, including STAT3 and HIF-1α genes [215]. Additionally, mannosylated pH-responsive nanoparticles were utilized to deliver siRNAs targeting placental growth factor (PIGF) and VEGF to TAMs. This approach allowed for specific and efficient delivery of the siRNAs to TAMs, resulting in the silencing of PIGF and VEGF genes [216]. In a study by Yu et al., a triblock copolymer was modified with mannose through click chemistry. The resulting mannose-functionalized copolymer was utilized to create a micellar system capable of delivering siRNA specifically to TAMs [217].

#### 6.2.3. Nanotechnology-Based Active Targeting Strategies

Nanotechnology-based targeting strategies have emerged as innovative approaches to specifically target TAMs within the tumor microenvironment. Because macrophages efficiently phagocytize NPs ranging from 0.1 nm to 1 μm, favoring elongated shapes and sharp edges, nanoscale materials and delivery systems could enhance the precision and efficacy of therapeutic interventions. Passive targeting delivery systems, exploiting the enhanced permeability and retention (EPR) effect, have been extensively studied to localize nanoparticles [218]. However, the efficacy of passive targeting is limited. Thus, an active targeting system is on demand. Through ligand-receptor interactions, active targeting facilitates selective accumulation of nanoparticles at target sites. Ideally, this approach exclusively interacts with the target cells, bypassing off-target toxicity concerns [219]. The representative works about the formulations targeting TAMs are presented in Table 2.

##### Nanobody

Tumor cells often overexpress PD-L1, leading to immune tolerance induction [220,221]. Liposome is a good formulation for targeting the cancer microenvironment [222,223]. PD-L1 can be used as a targeted site, as shown by Yin et al., with PD-L1 nanobody-encapsulated gefitinib and simvastatin liposomes to overcome EGFR T790M-associated drug resistance in non-small cell lung cancer (NSCLC) (Figure 7A) [224]. Immunoglobulin (Ig) has a Y-shaped structure with variable binding sites that can recognize specific regions on antigens. Mannose receptor-targeted nanobodies, derived from Camelidae heavy-chain antibodies, labeled with 99mTc, specifically label TAMs in the tumor microenvironment [225]. TAMs highly expressed CD163; based on this, enhanced TAM uptake can be achieved by ligating CD163-responsive monoclonal antibodies on pegylated liposomes [226]. A liposomal drug delivery system was developed to achieve dual targeting through the simultaneous modification with PD-L1 nanobody and mannose ligands. This system was designed for the co-delivery of an mTOR inhibitor (rapamycin) and an anti-angiogenic drug (regorafenib). The liposomes exhibited the capability to target both TAMs and cancer cells [182].

##### Carbohydrate Ligand

The mannose receptor CD206 can be effectively utilized for M2 TAM-specific targeting and potential delivery in cancer treatment [227]. The targeting ability of CD206 ligands-mannose has been confirmed by [217,228,229]. In another example, mannose moieties have been attached to pH-sensitive polymeric micelles using ‘click’ chemistry for siRNA encapsulation; with this modification, the gene silencing in TAM is up to 87 ± 10% [217]. In addition, a single-chain peptide binding the CD206 receptor conjugated to nanocarriers also allowed them to selectively target CD206 TAMs, even in hypoxic regions [225]. Oligomannose-coated liposomes (OMLs) loaded with 5-fluorouracil and magnetic NPs achieved a controlled release of 5-fluorouracil and improved tumor growth inhibition via peritoneal macrophage active internalization function [230]. To minimize uptake by normal macrophages, an acid-sensitive PEG modification in mannose-modified NPs was utilized, reducing uptake by the reticuloendothelial system (RES) through effective PEG shielding at neutral pH [231]. Additionally, for enhancing targeted gene delivery to alveolar macrophages, mannan-modified solid lipid nanoparticles (SLN) were investigated [232]. Other carbohydrate ligands such as dextran [233], galactosylated cationic dextran [234], and carboxydetran [235] are mainly used for imaging and will not be discussed here.

##### Proteins and Peptide

Albumin, the major protein in the blood, serves as a reservoir of amino acids [236]. Either tumor cells or TAMs have high demands for albumin as a source of amino acids. In the internalizing process, albumin-binding proteins, such as SPARC (secreted protein acidic and rich in cysteine), play a crucial role in the uptake of albumin by cells [237]. Combining the advantage of mannose and albumin, a dual-targeting mannosylated albumin nanoparticle was constructed; this nanoparticle targeting SPARC and MR (mannose receptor) can effectively target both cancer cells and M2 macrophages, leading to the reprogramming of the tumor microenvironment [229]. Furthermore, biomimetic albumin-modified gold nanorods (AuNRs) incorporating paclitaxel (PTX) have demonstrated tumor inhibition through the synergistic effects of photothermal and chemotherapy. Interestingly, AuNRs can also suppress the polarization of macrophages towards the M2 pro-tumor phenotype via SPARC-mediated uptake [198]. Surface modification of NPs with a minimal peptide corresponding to the CD47 receptor, a putative marker of self-recognition expressed in tumors, was also suggested to avoid the capture of NPs by RES phagocytosis [238].

##### Antibody

In addition to peptides, antibodies could also be utilized as ligands for targeted therapy.To combat EGFR mutation-related drug resistance, a novel approach involving a trastuzumab-modified, mannosylated liposomal system (tLGV) has been developed. This system targets HER2-positive NSCLC cells and mannose receptor (CD206)-overexpressed TAM2 simultaneously, co-delivering gefitinib and vorinostat. Although HER2-based treatments have been explored for NSCLC, HER2-targeted drug delivery in lung cancer has been scarcely investigated, with its feasibility yet to be demonstrated. This innovative approach holds promise as an alternative strategy to overcome EGFR mutation-associated therapeutic resistance in NSCLC vorinostat [239] (Figure 7B).

##### Legumain

Legumain, a cysteine protease belonging to the C13 family, is highly expressed in various solid tumors and TAMs [240]. Legumain is also highly expressed in a number of solid tumors [241,242]. As a result, it represents a promising therapeutic target. Importantly, because legumain is primarily expressed in M2 macrophages rather than classical M1 macrophages, legumain-mediated TAM targeting delivery does not induce cytotoxicity in normal M1 macrophages, nor does it interfere with the antigen presentation functions of M1 macrophages [243,244,245]. The use of legumain-expressing DNA vaccines stimulates a robust CD8+ T cell response against TAMs, leading to a significant reduction in TAM quantity within tumors. Consequently, this remodeling of the TME reduces its immunosuppressive properties, enhancing the effectiveness of the DNA vaccine in suppressing tumor growth and metastasis. The delivery of these therapeutic agents, or similar ones, using nanocarriers presents a novel and effective approach for targeted elimination of TAMs, leveraging the high specificity of drug-loaded nanoparticles for TAM uptake [246]. Specifically, legumain itself could be used as an adjuvant in cancer vaccines [247,248] (Figure 7C).

**Figure 7 pharmaceutics-16-00061-f007:**
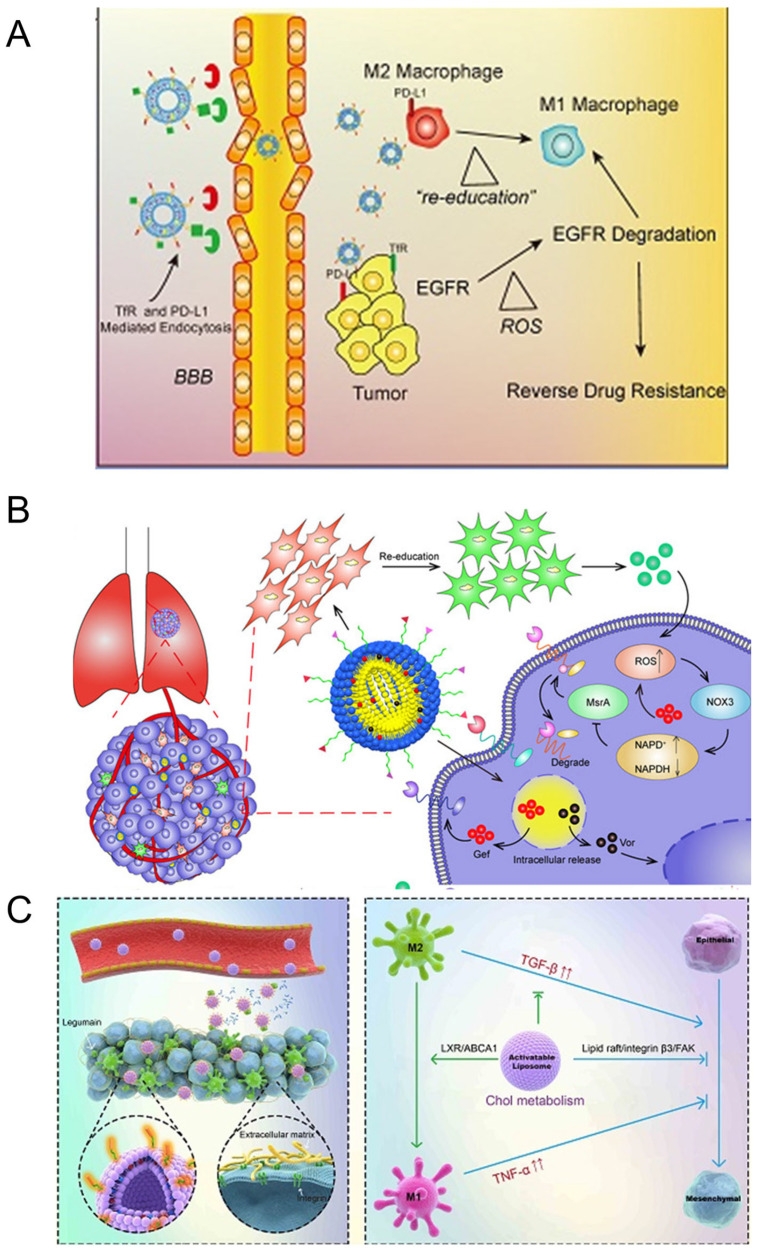
Nanotechnology-based strategies for reprogramming tumor microenvironment. (**A**) Remodeling tumor-associated macrophages and neovascularization overcomes EGFRT790M-associated drug resistance by PD-L1 nanobody-mediated codelivery of Gefitinib and Simvastatin [224]. Copyright © 2018 WILEY-VCH Verlag GmbH & Co. KGaA, Weinheim, Germany. (**B**) Reprogramming tumor-associated macrophages to reverse EGFRT790M resistance by dual-targeting codelivery of gefitinib/vorinostat [239]. Copyright © 2017 American Chemical Society (↑ suggesting the increase, ↓ suggest the decrease). (**C**) Targeting lipid metabolism to overcome EMT-associated drug resistance via integrin β3/FAK pathway and tumor-associated macrophage repolarization using legumain-activatable delivery [249]. This is an open-access article distributed under the terms of the Creative Commons Attribution License.

**Table 2 pharmaceutics-16-00061-t002:** Summary of recent developments in macrophage-mediated/related drug delivery and treatments.

Modified Ligand or Formulation	Modality	Reference
Macrophage-Membrane-Coated Nanoparticle	A macrophage-membrane-coated nanoparticle (cskc-PPiP/PTX@Ma) has been engineered for targeted chemotherapy delivery with controlled release in response to tumor microenvironment stimuli. After tumor homing and evasion of the reticuloendothelial system, the macrophage-membrane coating undergoes morphological changes triggered by extracellular stimuli, facilitating the shedding of nanoparticles. This biomimetic drug delivery system, combining membrane-derived tumor homing and step-by-step controlled drug release, demonstrates enhanced therapeutic efficacy tailored to the intricacies of the tumor microenvironment.	[250]
PD-L1 nanobody liposome	Reprogramming TME for overcoming drug resistance.	[224]
Carbohydrate-Functionalized Polymeric Nanoparticles	A review illustrates that carbohydrate-modified nanoparticles are able to modulate macrophages M1/M2 polarization.	[251]
Hemoglobin-decorated liposomes	A sophisticated drug delivery system where the integration of hemoglobin onto the liposomal surface enhances stability and may potentially facilitate oxygen transport, offering a multifaceted platform for targeted therapy. Specifically targeting macrophages through the CD163 receptor.	[252]
Mannose-NPs	By delivering therapeutic agents directly to TAMs, Mannose-NPs can modulate the tumor immune response, suppress pro-tumorigenic activities of TAMs, and ultimately impede cancer progression, offering a promising strategy for precision cancer therapy.	[217,229,230,231,238]
Mannosylated polymeric micelles	Mannosylated polymeric micelles, employing “click” chemistry, offer a targeted delivery platform for RNA interference molecules to macrophages by utilizing the overexpressed mannose receptors, providing a precise and efficient means of modulating gene expression in immune cells.	[252]
Dextran-labelled zirconium-89 (89Zr)	The development involves a novel macrophage-specific positron emission tomography (PET) imaging agent labeled with zirconium-89 (89Zr). This agent is constructed using a cross-linked, short-chain dextran nanoparticle with a size of 13 nanometers. After systemic administration, the nanoparticle exhibits a vascular half-life of 3.9 h, and notably, it preferentially accumulates in tissue-resident macrophages as opposed to other white blood cells.	[253]
Surface decoration of delivery vectors	The surface decoration of delivery vectors, despite not influencing the expression of M1 markers such as CD86, NOS2, TNF-α, and IL-1β, demonstrated a distinct impact on the expression levels of scavenger receptors CD163 and CD200R, as well as the release of the anti-inflammatory cytokine IL-10.	[254,255]

## 7. Summary and Conclusions

In summary, this review presented the role of TAMs in the tumor immune microenvironment and how TAMs are involved in metastasis, angiogenesis, lymphangiogenesis, metastatic colonization, survival, and therapeutic resistance. This article discussed conventional and nanoparticle-based therapeutic strategies and the use of nanoparticles for imaging TAMs and delivering therapeutics, encompassing MRI-visible nanoparticles, and large molecules therapeutics.

TAMs-based therapies play a complex role in tumor immunity and immunotherapy via modulation of M1 and M2 phenotypes. However, achieving a delicate balance between M1 and M2 macrophages is essential for the safety of immunotherapy, as an excess of M1 macrophages may lead to adverse effects like cytokine release syndrome [256]. Monitoring cytokine levels or using advanced organ-on-chip platforms for prediction becomes pivotal to minimizing unnecessary organ damage during TAM therapies.

Moreover, the complex and diverse nature of M2 TAM subpopulations in animals and humans poses a significant problem in developing TAM-targeted nanoparticles to attain desired therapeutic results. While certain biomarkers like CD206 and CD163 can accurately detect TAM populations, it remains challenging to ascertain their specificity as CD206, for instance, is also present in liver cells [257]. Hence, identifying reliable biomarkers is crucial in understanding the role of TAMs in cancer.

The above issues pose challenges to efficient delivery and specific targeting of TAM. In addition, TAM distribution can vary between tumors and even within different regions of the same tumor [258]. Though we explore the therapeutic potential of using nanoparticles, the accuracy of EPR effects in humans is challenging. Therefore, developing strategies to effectively deliver therapeutic agents to TAMs while minimizing off-target effects such as liver have been identified as a significant issue for TAM therapies.

In conclusion, it is essential to understand the potential side effects and toxicities associated with TAM-targeted therapies. More research is required to address the challenges associated with TAM-targeted therapies. Many modalities have been attempted to develop effective TAM-based cancer treatment. This approach requires more research in the future to elucidate the role of nanoparticle-based systems on TAM biology and associated biomarker identification.

## Figures and Tables

**Figure 1 pharmaceutics-16-00061-f001:**
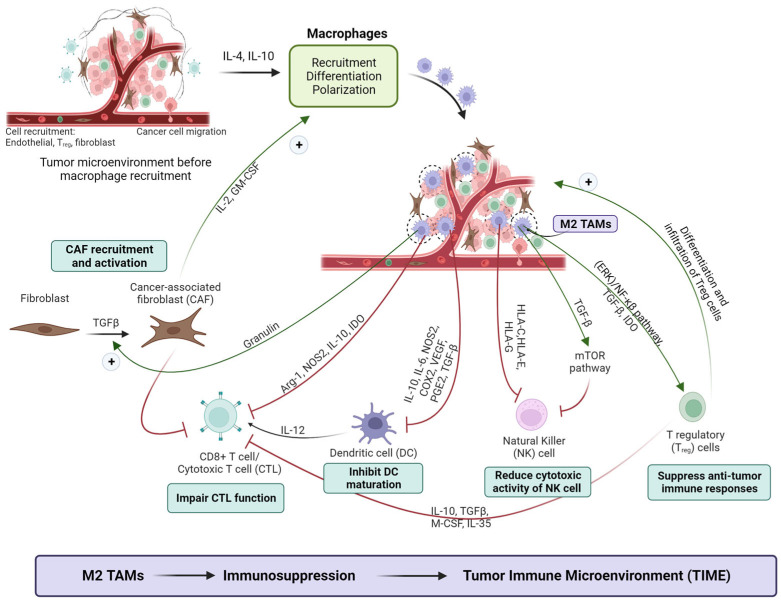
In the tumor niche, tumor cells release MCSF, IL4, and IL10, etc., to attract macrophage; then, tumor-associated macrophages engage in intricate interactions with cancer-associated fibroblasts (CAFs), T regulatory cells, natural killer cells to form an immunosuppressive tumor microenvironment (TME). Understanding these interactions is crucial for developing targeted therapies to overcome immunosuppression in the TME. (Images created with biorender.com, accessed on 17 July 2023).

**Figure 2 pharmaceutics-16-00061-f002:**
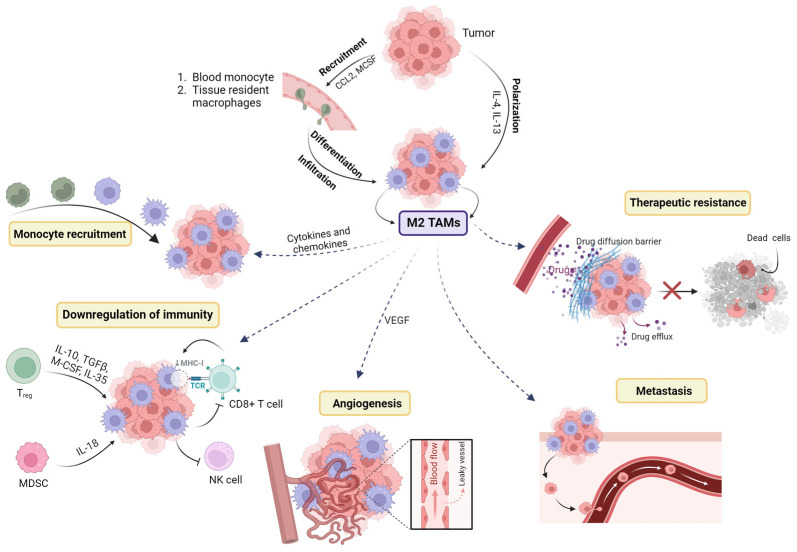
The role of macrophages in TME: In the context of tumor progression, neoplastic cells and stromal cells release specific molecules that act as chemoattractants, such as CCL2 and MCSF-1, to recruit circulating monocytes to the tumor site. Once recruited, monocytes significantly differentiate into M2 TAMs. The predominant M2 TAMs promote the downregulation of tumor immunity, angiogenesis, as well as therapeutic resistance (↓ suggesting decrease) (created with biorender.com, accessed on 17 July 2023).

**Figure 3 pharmaceutics-16-00061-f003:**
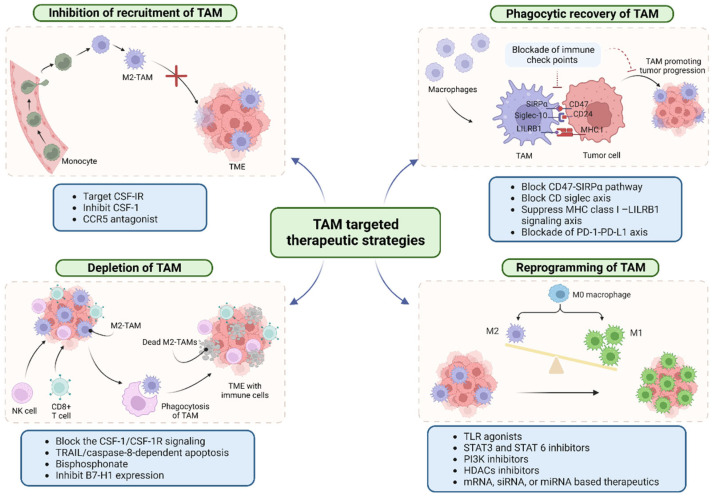
Schematic representation of a variety of therapeutic approaches targeting TAM. Targeting TAMs therapeutic strategies involve inhibiting TAM recruitment and differentiation, depleting or impairing their function, reprogramming M2 TAMs, and promoting their phagocytic activity. Inhibiting TAM recruitment entails blocking chemokine and growth factor signaling while inhibiting TAM differentiation involves targeting factors like IL-4 and IL-13. Depleting TAMs can be achieved through selective elimination using specific markers or immunotherapies. Impairing TAM function targets signaling pathways involved in immunosuppression and angiogenesis. Reprogramming M2 TAMs toward an anti-tumoral M1-like phenotype enhances their anti-tumor activity. Promoting TAM phagocytosis enhances their ability to eliminate tumor cells. Combination therapies, integrating TAM-targeting approaches with other modalities, hold promise for synergistic effects (Created with biorender.com, accessed on 17 July 2023).

**Figure 4 pharmaceutics-16-00061-f004:**
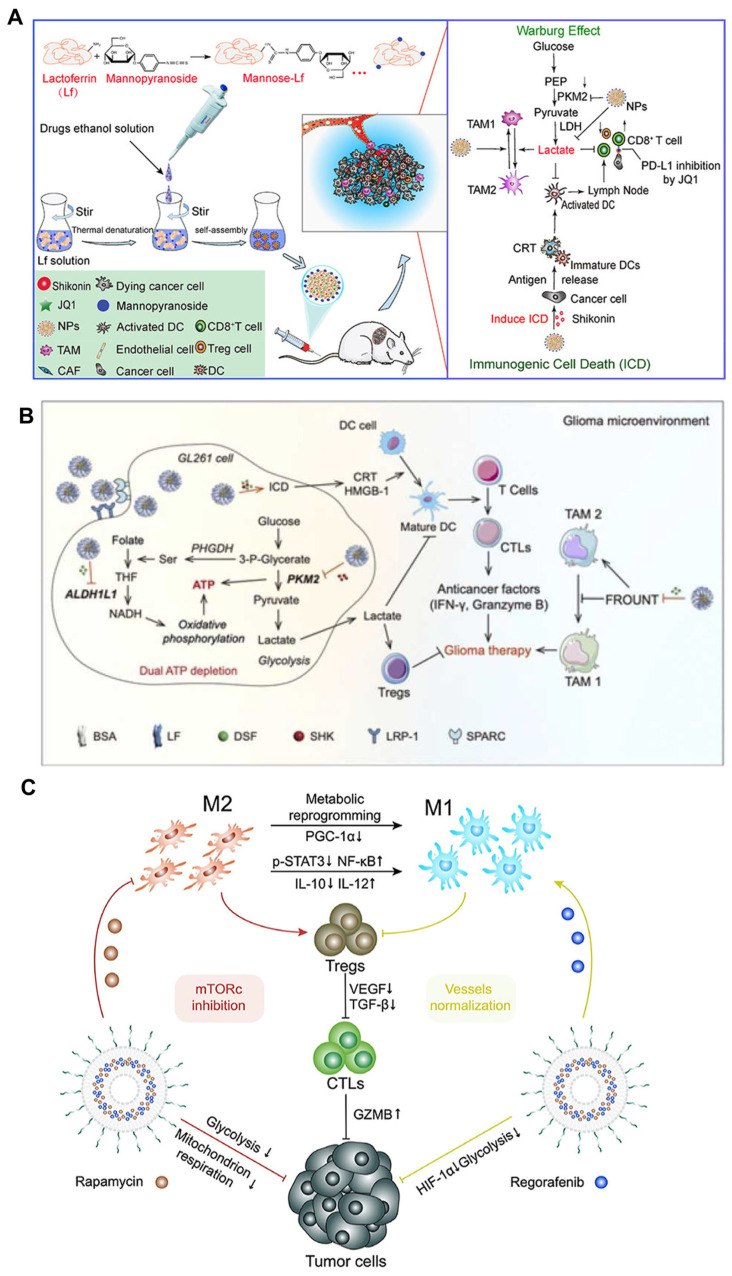
Represenative of therapeutic modality for metabolism in TAM. (**A**) Schematic illustration of biomimetic targeting codelivery of Shikonin/JQ1 for reprogramming TME via regulation of metabolism (↑ suggesting increase, ↓ suggesting decrease) [180]. Copyright © 2019 American Chemical Society. (**B**) Schematic illustration of anti-alcoholism drug disulfiram for targeting glioma energy metabolism [178]. Copyright © 2022 Published by Elsevier Ltd. (**C**) Schematic illustration of using a PD-L1-targeting system loaded with rapamycin and regorafenib for metabolic modulation in TME [182]. Copyright © 2020 Elsevier Ltd.

**Figure 5 pharmaceutics-16-00061-f005:**
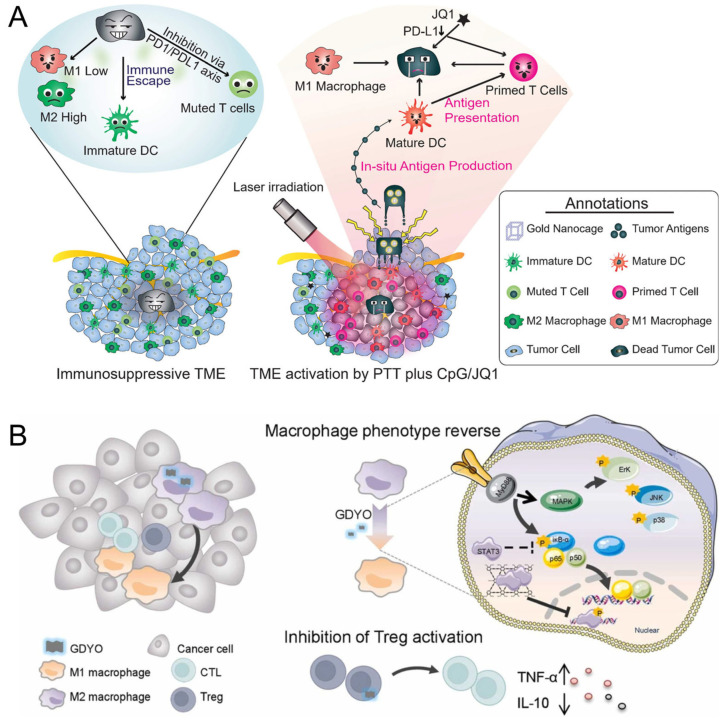
Representative work of inorganic nanomaterials for TAM modality. (**A**) Schematic illustration of AuNC-based in situ vaccination: the photothermal tumor ablation of AuNC cut the source of TAM differentiation; the combination of AuNCs with the JQ1 (PD-L1 suppressor) dramatically inhibit the function of M2 TAM that overexpress PD-L1 [197]. Copyright © 2022 American Chemical Society. (**B**) Graphdiyne oxide nanosheets reprogram immunosuppressive macrophages for cancer immunotherapy (↑ suggesting the increase, ↓ suggesting the decrease) [204]. Copyright © 2022 Elsevier Ltd.

**Figure 6 pharmaceutics-16-00061-f006:**
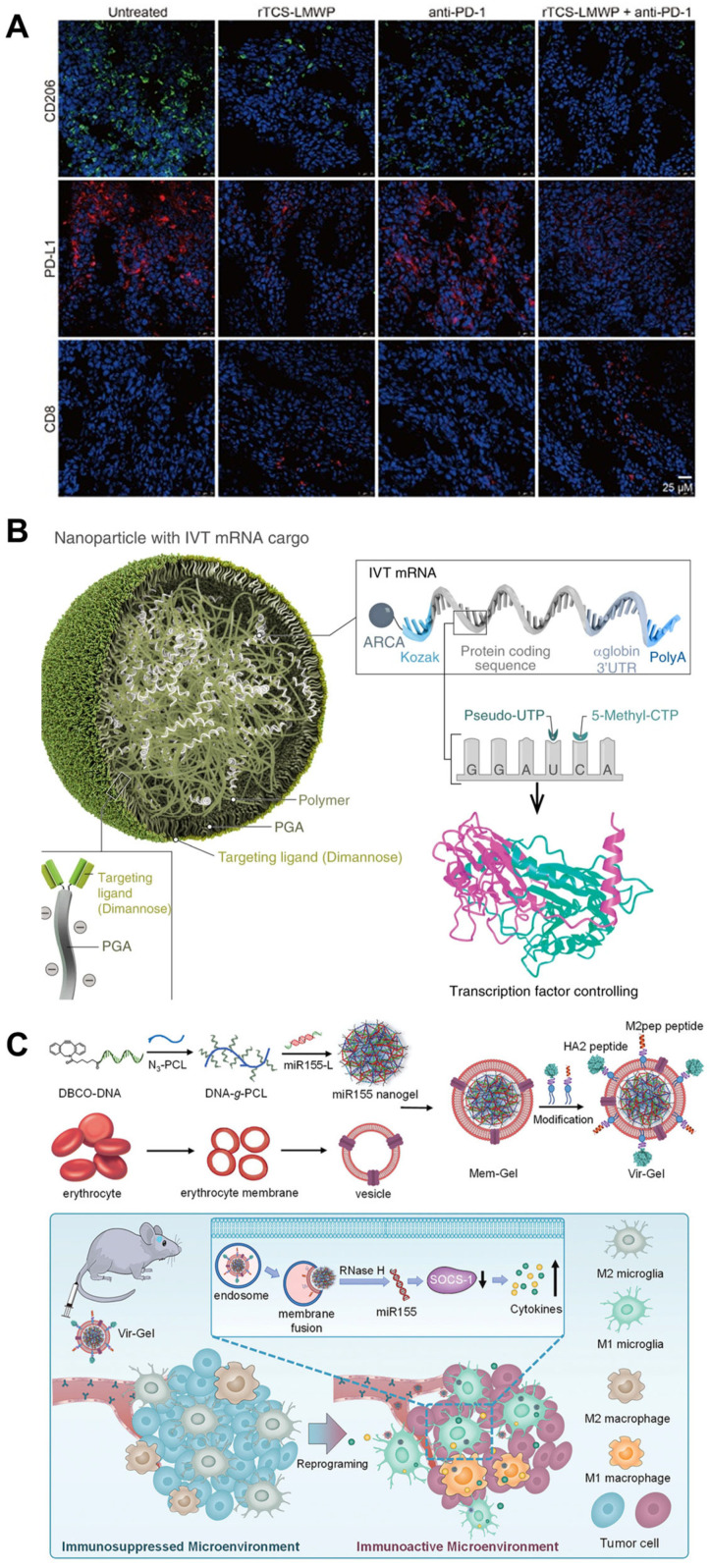
Representative work of inorganic nanomaterials for TAM modality. (**A**) Schematic representation of recombinant cell-penetrating trichosanthin synergizes anti-PD-1 therapy in colorectal tumor [208]. This is an open-access article distributed under the terms of the Creative Commons Attribution License. (**B**) Schematic representation of using mannose-modified system for mRNA delivery. Reproduced from reference [210]. Open-access article distributed under the terms of the Creative Commons CC BY license. (**C**) Schematic representation of using erythrocyte membrane-coated virus-mimicking nanogel for miRNA delivery [210]. Copyright © 2021 John Wiley and Sons.

**Table 1 pharmaceutics-16-00061-t001:** Summary of therapeutics targeting TAM.

Target Site	Substance	Ref or Identifier
CSF1/CSF1R	AMG820 BLZ945 PLX3397 PLX7486 GW2580 RG7155 Cabiralizumab(FPA008) IMC-CS4	NCT01444404 [117] NCT02829723 [118] NCT01349049 [119] NCT01804530 [120] NCT01494688 [121] NCT03336216 NCT01346358, NCT02265536, NCT03153410
CCL2/CCR2	Carlumab MLN1202 PF04136309 RS102895 Zoledronic acid	NCT00992186 [122] NCT01015560, NCT01413022 NCT02732938 [123,124] [106]
CCL5/CCR5	Maraviroc Vicriviroc Gefitinib	[125] [126] [127]
PD-1/PDL-1	CD3-HAC/ BMS-936558 PD-1 antibody	[128] [26] [129]
CD47-SIRPα	Hu5F9-G4 Glutaminyl cyclase	NCT02953509, NCT02953782 [130] [131]
B7-H1	Amphotericin B	[132]
PI3K	Wortmannin or LY294002	[133]
RKIP	siRNA	[134]
AMP-activated protein kinase (AMPK)	RSVA314, RSVA405	[135]
STAT3	CPA-7, AZD9150 WP1066	[136] NCT03421353 [137] NCT01904123 [138]
Leukocyte Ig-like receptor (LIR) 1 (CD85j/ILT2/LILRB1)	GHI/75 (nti-LIR-1 antibodies)	[139]
CD24/Siglec-10	Genetic ablation or Monoclonal antibodies	[140]
Arg-1	Cyclosporine	[141]
TLR7	852A Imiquimod	[142] [143,144]
TLR7/8	Resiquimod(R848)	[109]
TAM apoptosis	Clodronate	[145]
	Trabectedin	[146]
	Bisphosphonate	[147]

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
