# Peer review of "Tumor-Associated Macrophage Targeting of Nanomedicines in Cancer Therapy"

_pharmaceutics, 2023, doi:10.3390/pharmaceutics16010061_

Round 1
Reviewer 1 Report
Comments and Suggestions for Authors
Dear colleagues,
In this manuscript, the authors perform a review of the participation of tumor-associated macrophages in oncologic processes and possibility to influent of its activity with nanoparticles. The manuscript is interesting. The figures reflect the aim of the work. Despite the very good impression of the article, there are some questions which could improve the article in my opinion, partly:
There are numerous technical mistakes with missed space, absent space, dot before square brackets, and others.
The final resume on the conclusion should be more concrete and should be expanded.
In summary, I have been satisfied with the high level of the article. I believe this manuscript will attract significant attention from the research community. In my personal opinion, the article is very valuable, a great prospect for further research, and, after minor corrections, can be recommended for publication.
Author Response
Please see attached response

Reviewer 2 Report
Comments and Suggestions for Authors
The manuscript entitled “Nanoparticle-Mediated Modulation of Tumor-Associated Macrophages: A Frontier in Cancer Therapy” deals with innovative nanoparticle-based approaches in tumor management. This narrative review is a valuable and comprehensive summary of the results of previously published articles on this issue, the topic of the manuscript is interesting to the readership focusing on cancer biology and immunology and it is within the scope of this journal.
The manuscript is very well written and well structured, while the overall presentation is clear and informative. There are merely some minor issues that need to be addressed:
- These is some extra text in lines 64 and 65 (“reprogramming M2 TAMs”).
- In line 52 there are some extra parentheses in reference numbering.
Author Response
Please see attached response

Reviewer 3 Report
Comments and Suggestions for Authors
Review on the manuscript “Nanoparticle-Mediated Modulation of Tumor-Associated Macrophages: A Frontier in Cancer Therapy” by Xuejia Kang et al.
According with the literature data, in cancer therapy, targeting tumor-associated macrophages has shown promise. In addition, the control over the tumor-associated macrophages with the help of nanoparticles led to improved therapeutic effect.
This review is focused on the tumor-associated macrophages capability to influence the tumor microenvironment and on the effect of the nanoparticles over the tumor-associated macrophages; even do the title underlines the “nanoparticle-mediated modulation” of tumor-associated macrophages. Maybe the authors should reconsider the title or the review’s objectives.
The review is interesting but I have some questions that I think the authors should consider:
The abstract should underline the main finding of this review.
What protocol was used when were selected the studies included in this review?
Please present the databases or other resources used to identify the studies. Were used some criteria or limitations in this process? How were these studies selected?
Did the authors consulted the PRISMA guidelines?
The introduction is to short, has nearly a page. All the sections until section 6 should be reorganized as subsections of the introduction.
Starting with section 6 is practically presented the nanoparticle influence on tumor-associated macrophages.
Please use the same abbreviation in the manuscript, for example M2 TAMs, m2 TAMs or M2TAMs.
Some of the figures are practically illegible, please carefully verify them.
What are the limitations of the review?
Comments on the Quality of English LanguagePlease carefully check the manuscript for misspellings.
Author Response
Please see attached response

Reviewer 4 Report
Comments and Suggestions for Authors
Dear Authors,
Re: pharmaceutics-2761167
Title: Nanoparticle-Mediated Modulation of Tumor-Associated Macrophages: A Frontier in Cancer Therapy
Your article, which is in the form of Review article, aims to cover challenges and limitations of the targeted delivery of drugs using tumor microenvironment and TAMs that is expected to become a more practical approach for more efficacious cancer nanotherapy. I enjoyed reading your review article. Please check the following suggestions / comments:
In the Abstract, please double check this sentence:
"... This review underscores the critical roles TAMs play in the tumor microenvironment and explores the ..."
In Lines 59, 73 and 76 (as well as few more places) please "leave a space" between last word and citations.
Only 7 of the 270 References belong to 2023. You can incorporate more recent literature in your article. For instance:
https://doi.org/10.3390/pharmaceutics15071920
and https://doi.org/10.1016/j.pdpdt.2023.103614
Note that "nanoliposomes" are one of the top strategies with potential for tumor targeting and this strategy is missing in your paper.
Another recent and highly important relevant topic is cancer photodynamic therapy. This concept can be mentioned in a couple of lines using 2023 References e.g.:
https://doi.org/10.3390/biomedicines11010224
The topic of "nanobody" is not described adequately in the section (Lines 522-531).
Figures texts are too small and cannot be seen properly.
Any copyright issues with the Figures?
Please separate Figure 1 legend and the text.
In my opinion, EPR deserves more explanations.
The Table of Abbreviations need Table legend.
Thank you.
Comments on the Quality of English LanguageEnglish language of the manuscript is fine. Just few punctuation errors detected.
For instance:
In Lines 59, 73 and 76 (as well as few more places) need to "leave a space" between last word and citations.
Author Response
Please see attached response
